# Changes following the Onset of the COVID-19 Pandemic in the Burden of Hospitalization for Respiratory Syncytial Virus Acute Lower Respiratory Infection in Children under Two Years: A Retrospective Study from Croatia

**DOI:** 10.3390/v14122746

**Published:** 2022-12-09

**Authors:** Dina Mrcela, Josko Markic, Chenkai Zhao, Daniela Veljacic Viskovic, Petra Milic, Roko Copac, You Li

**Affiliations:** 1School of Medicine, University of Split, Soltanska 2, 21000 Split, Croatia; 2Department of Pediatrics, University Hospital of Split, Spinciceva 1, 21000 Split, Croatia; 3Department of Epidemiology, School of Public Health, Nanjing Medical University, 101 Longmian Avenue, Jiangning District, Nanjing 211166, China; 4Department of Physics, Faculty of Science, University of Split, Rudera Boskovića 33, 21000 Split, Croatia

**Keywords:** respiratory syncytial virus, COVID-19, children, bronchiolitis, severity

## Abstract

To understand the changes in RSV hospitalization burden in children younger than two years following the onset of the COVID-19 pandemic, we reviewed hospital records of children with acute lower respiratory infection (ALRI) between January 2018 and June 2022 in Split-Dalmatia County, Croatia. We compared RSV activity, age-specific annualized hospitalization rate, and disease severity between pre-COVID-19 and COVID-19 periods. A total of 942 ALRI hospital admissions were included. RSV activity remained low for the typical RSV epidemic during 2020–2021 winter. An out-of-season RSV resurgence was observed in late spring and summer of 2021. Before the COVID-19 pandemic, the annualized hospitalization rate for RSV-associated ALRI was 13.84/1000 (95% CI: 12.11–15.76) and highest among infants under six months. After the resurgence of RSV in the second half of 2021, the annualized hospitalization rate for RSV-associated ALRI in children younger than two years returned to the pre-pandemic levels with similar age distribution but a statistically higher proportion of severe cases. RSV immunization programs targeting protection of infants under six months of age are expected to remain impactful, although the optimal timing of administration would depend on RSV seasonality that has not yet been established in the study setting since the onset of the COVID-19 pandemic.

## 1. Introduction

Acute respiratory tract infection (ALRI) represents a substantial burden to the health system and remains one of the main causes of mortality and morbidity in young children [1,2]. Respiratory syncytial virus (RSV) is the predominant respiratory pathogen found in children under five years with ALRI [3]. Globally in 2019, there were 33 million RSV-associated ALRI episodes and 3.6 million hospitalizations, causing over 100,000 deaths; a disproportionally higher burden occurred in infants under six months [4]. RSV has distinct seasonal circulating patterns in most parts of the world, lasting about 5 months in each season, although the exact timing of circulation varies by latitude among other factors [5,6,7,8].

Substantial investments have been put into the development of a RSV vaccine and immunoprophylaxis in the recent five years to protect at-risk populations, especially infants aged under one year. The RSV prophylactic product furthest in the pipeline has shown promise in granting protection for five months for infants aged 6 months or younger throughout a typical RSV season [9]. However, following the onset of the COVID-19 pandemic, the circulating patterns of various respiratory pathogens including RSV have been impacted as a result of public health measures, with historically low activity early on followed by out-of-season resurgence across the world [10,11,12,13,14]. In addition to the timing of RSV circulation, there were also studies highlighting changes in age distribution and disease severity of RSV-associated ALRI in young children compared with the pre-pandemic period, although the reported changes were not consistent across different studies [15,16,17,18]. These changes in RSV circulation timing, age distribution, and disease severity in the COVID-19 pandemic era pose challenges to RSV immunization planning for infants as to whether the timing and the targeted population (i.e., currently being primarily those under six months) need to be reconsidered.

Here, we aimed to understand the potential changes in hospital admission burden of RSV-associated ALRI in children under two years following the onset of the COVID-19 pandemic with regard to circulating timing, age distribution, and disease severity.

## 2. Materials and Methods

### 2.1. Participants

This was a retrospective study conducted at the Department of Pediatrics of the University Hospital of Split. Patients under the age of two years who were admitted for at least 24 h between 1 January 2018 and 30 June 2022 and diagnosed with ALRI (bronchitis, bronchiolitis, pneumonia) were included. All relevant medical records including any laboratory testing records were extracted for analysis. Ethical approval was granted by the Ethics Committee of the University Hospital of Split (No. 2181-147/01/06/MS-22-04).

### 2.2. Definitions

A subset of ALRI was defined as severe and very severe ALRI based on the criteria below: severe ALRI was defined as ALRI with hypoxemia (defined as peripheral capillary oxygen saturation, SpO_2_ < 90%), and very severe ALRI was defined as ALRI with at least one of the following signs, cyanosis, difficulty in feeding or drinking, inability to consume sufficient food and liquids to meet nutritional and hydration requirements, convulsions, lethargy, or unconsciousness, severe dyspnea, and requirement for mechanical ventilation or intensive care unit (ICU) admission. Figure 1 illustrates the roadmap of implantation of non-pharmaceutical intervention in Croatia. We define the onset of the COVID-19 pandemic based on the local context, that is 16 March 2020, when lockdown was imposed in Croatia. Therefore, the period of 1 January 2018 to 16 March 2020 was defined as the pre-COVID-19 period, and 17 March 2020 to 30 June 2022 was defined as the COVID-19 period.

### 2.3. Testing for RSV

Testing for RSV was ordered at the discretion of the physicians and testing practices were unchanged during the whole study period. Nasal aspirate specimens were collected from the patients immediately upon admission and tested using rapid antigen test (RSV-AdenoGnost RESP, BioGnost, Zegrab, Croatia) on the same day (or the following morning for midnight admission). All children admitted to the hospital during the COVID-19 period were simultaneously tested for SARS-CoV-2 as well, when testing became available.

### 2.4. Data Analysis

Subject characteristics including age, sex, gestational age, birth weight and height, comorbidities, and severity of ALRI were compared between pre-COVID-19 and COVID-19 periods separately for all-cause ALRI hospital admissions and for RSV-associated ALRI hospital admissions. These comparisons were done by applying the Chi square test for categorical variables (or Fisher’s exact test where deemed necessary) and the Mann–Whitney U test for continuous variables.

To understand the temporal trend of burden of hospital admission, we calculated the 12-month moving average hospitalization rates of all-cause ALRI and RSV-associated ALRI in different age groups (0–<6 months, 6–<12 months, and 12–<24 months). For the pre-COVID-19 period, this started with estimating the 12-month hospitalization rate for January 2018 to December 2018, and ended with March 2019 to February 2020. For the COVID-19 period, this started with March 2020 to February 2021 and ended with July 2021 to June 2022. The calculated 12-month hospitalization rate reflected the average burden of hospitalization in the given period that accounts for annual RSV seasonality. Similar to our previous work, the hospitalization rate of RSV-associated ALRI was adjusted by accounting for under-testing of RSV with the assumption that the RSV positivity rate among ALRI not being tested for RSV was equal to that among ALRI being tested for RSV, for each age group per calendar month [19]. The population denominator for the rate calculation was extracted from the national yearbook [20].

To further understand whether severity of ALRI was associated with the onset of the COVID-19 pandemic, we conducted multivariate logistic regression of severity of all-cause ALRI and RSV-associated ALRI (defined as any severe or very severe ALRI) with the following predefined independent variables deemed to be associated with ALRI severity: sex, age group (0–<6 months, 6–<12 months, and 12–<24 months), prematurity (defined as gestational age of <37 weeks), low birth weight (defined as <2500 g), presence of any comorbidities, and whether the admission was after the COVID-19 pandemic. We further conducted a stratified analysis by age group to explore whether any observed associations were consistent across different age groups. As sensitivity analysis, we replaced the independent variable of presence of any comorbidities with the number of comorbidities, and with individual comorbidities as multiple independent variables.

## 3. Results

Between 1 January 2018 and 30 June 2022, a total of 942 patients fulfilled the inclusion criteria and were included in the study. Out of them, 592 (62.8%) were admitted before 16 March 2020 when lockdown was imposed in Croatia and 350 (37.1%) during the COVID-19 pandemic. A total of 695 (73.8%) of ALRI hospitalizations were tested for RSV. Demographic and clinical characteristics of study subjects with all cause ALRI and RSV-associated ALRI before and during the COVID-19 period are summarized in Table 1. Compared with the pre-COVID-19 period, the proportion of severe and very severe cases as well as length of stay was statistically higher in COVID-19 period, for both all-cause and RSV-associated ALRI hospital admissions (Table 1). In addition, for all-cause ALRI hospital admissions, the age distribution was different between pre-COVID-19 and COVID-19 periods; the presence of any comorbidities and the presence of congenital heart disease were more frequently found during the COVID-19 period; the birth height of those admitted with ALRI was also statistically significant lower during the COVID-19 period than the pre-COVID-19 period (Table 1).

Regarding testing practice, ALRI cases that were under six months and born prematurely were more likely to be tested for RSV (Appendix A); compared with the pre-COVID-19 period, ALRI cases were more likely to be tested for RSV during the COVID-19 period (Appendix A). Compared with those receiving negative RSV test results, testing positive for RSV was more often found in younger children and those without comorbidities (Appendix A).

In the pre-COVID-19 period, ALRI admissions peaked in the winter months, with the highest proportions of RSV-associated admissions typically seen between December and April, whereas during the summer, there were no RSV cases recorded. Since the national lockdown in March 2020, both all-cause and RSV-associated ALRI admissions dropped steadily, and the number remained low throughout the 2020–2021 winter months. An out-of-season RSV outbreak was observed during late spring and the summer months of 2021; in the following autumn months, RSV-associated ALRI admissions increased rapidly, peaked in October 2021, and then declined sharply after January 2022 (Figure 2).

Before the COVID-19 pandemic, the annualized hospitalization rates among children under 24 months were 28.97/1000 (95% CI: 26.44–31.69) for all-cause ALRI and 13.84/1000 (12.11–15.76) for RSV-associated ALRI; the highest admission rate was found in infants under six months for both all-cause and RSV-associated ALRI. During the pre-COVID period, there was a long-term decreasing trend in the 12-month hospitalization rates for both all-cause and RSV-associated ALRI. During the COVID-19 pandemic, the 12-month hospitalization rates remained at a low level for the year 2020 but began constantly increasing as a result of RSV resurgence in the second half of 2021; the 12-month hospitalization rates for both all-cause and RSV-associated ALRI returned to comparable levels with those of the pre-COVID-19 period (Figure 3). However, the rates were consistently much higher in the COVID-19 period (after the resurgence of RSV in 2021) than the pre-COVID-19 period when restricting to only severe and very severe ALRI (Appendix A).

To better understand the association between the timing of ALRI hospital admission relative to the COVID-19 pandemic with disease severity, a series of multivariate logistic regression analyses were performed for all-cause of ALRI and RSV-associated ALRI, separately. For both all-cause and RSV-associated ALRI, admission during the COVID-19 pandemic tended to be more severe after adjustment for a range of potential confounders, with estimated ORs of 1.92 (1.17–3.15) and 5.03 (2.01–13.89), respectively (Table 2).

A stratified analysis by three age groups showed that for all-cause ALRI, the OR of admission during the COVID-19 period for more severe cases was highest in children in their second year of life, followed by infants under six months, and was not statistically significant for infants older than six months and younger than one year (Appendix A); for RSV-associated ALRI, the OR was as high as 7.65 (2.54–28.69; Appendix A). Sensitivity analysis that used different forms of variables for comorbidity status did not yield any substantial differences from the main analysis (Appendix A).

## 4. Discussion

In the present study, we report the changes following the onset of the COVID-19 pandemic in the burden of hospitalization for all-cause and RSV-associated ALRI in children under two years residing in Split-Dalmatia County, Croatia. We show that both all-cause and RSV-associated ALRI hospital admissions dropped dramatically following the national lockdown of Croatia as well as the strict implementation of public health measures such as wearing masks, enhanced hand hygiene, stay-at-home orders, and reduction of physical gatherings. The number of all-cause and RSV-associated ALRI hospital admissions remained low during the typical seasonal epidemic months in 2020–2021 winter until an out-of-season RSV outbreak occurred in late spring and summer in 2021. Annualized hospital rates for both all-cause and RSV-associated ALRI returned to pre-COVID-19 levels after 2021 but with a greater proportion of severe cases. Multivariate regression analysis confirmed that the ALRI admission during the COVID-19 period tended to be more severe after adjusting for potential confounders such as age and comorbidities.

The overall trajectory of RSV activity following the onset of the COVID-19 pandemic in our study site was broadly similar to other parts of the world, where RSV activity remained low initially and surged, often out-of-season, following the relaxation of NPIs [14,18]. Interestingly, RSV outbreaks were not observed immediately after the relaxation of NPIs, suggesting that relaxation of NPIs alone was not sufficient for a resurgence of RSV. A multi-country analysis suggested that the accumulation of a susceptible population was an important driver for out-of-season outbreaks that could override the effect of meteorological factors. Moreover, viral interference might also have played an important role [21]. In our study site, RSV seasonality did not precisely return to its pre-pandemic timings, and it remains unknown whether, with minimal public health measures, RSV seasonality could follow its pre-pandemic timings in the 2022–2023 season.

Several studies highlighted the older median age of RSV infections during the COVID-19 period, likely due to the immunity debt as a result of reduced exposure to RSV during the first expected RSV season following the onset of the COVID-19 pandemic [16,22,23]. However, in this study, we did not observe statistically significant differences in the age distribution between pre-COVID-19 and COVID-19 periods, which was also observed in a recent study from Israel [24]. Our data show that infants under six months remain the age group with the highest RSV hospitalization burden during the COVID-19 period and should be prioritized for preventive interventions such as RSV monoclonal antibodies or maternal vaccines. We further show that the absolute RSV hospitalization burden measured by 12-month hospitalization rate had returned to pre-pandemic levels since the second half of 2021, confirming the value for the investment of novel RSV prophylactic products despite the impact of the COVID-19 pandemic.

Regarding disease severity, previous studies reported mixed findings on whether RSV disease was more severe during the COVID-19 period [16,23,25,26]. The mixed findings were likely a result of different RSV epidemic trajectories that different countries underwent. For example, the population immunity levels to RSV might differ by countries at the beginning of the COVID-19 pandemic, which would further differentiate over time given the variations in the timing and composition of public health measures by countries. In this study, we demonstrate a higher proportion of severe and very severe all-cause and RSV-associated ALRI admissions during the COVID-19 period, after adjusting for potential confounders such as age and comorbidities; it is unknown whether the observed increased severity will persist in the following RSV seasons.

We acknowledge several limitations of our study. First, our analysis was based on a retrospective data collection of hospital records and therefore, could be subject to selection bias in terms of testing practice for RSV and information bias in terms of consistencies in clinical practice over time and among different clinicians, especially during the early phase of the COVID-19 period, when the indications for hospitalization might be stricter than before. Specifically, about a quarter of the ALRI admissions were not tested for RSV, and testing for RSV was more frequently done during the COVID-19 period compared with the pre-COVID-19 period. To account for non-testing for RSV, we adjusted for the RSV-associated ALRI hospitalization rate for each age group per calendar month as done in the previous analysis [19]. In addition, rapid antigen tests were used in our study setting, and this could result in the underestimation of the RSV-associated hospitalization burden. Second, we did not include other respiratory viruses than RSV in this analysis, which could help explain the potential role of viral interference in shaping RSV out-of-season epidemics. Third, our study was based on a single center representing the county of Croatia, and the findings might not be representative of the entire country; a recent study from Australia highlighted substantial subnational differences in the RSV epidemics during the COVID-19 period [27].

## 5. Conclusions

Following the initial low activity in the first winter since the COVID-19 pandemic, the hospitalization burden of RSV-associated ALRI in children younger than two years returned to pre-pandemic levels after mid-2021 with similar age distribution but increased disease severity. However, these findings, especially regarding disease severity, warrant further research. RSV immunization programs targeting protection for infants under six months of age are expected to be impactful, although the optimal timing of administration would depend on RSV seasonality that has not yet been established since the onset of the COVID-19 pandemic.

## Figures and Tables

**Figure 1 viruses-14-02746-f001:**
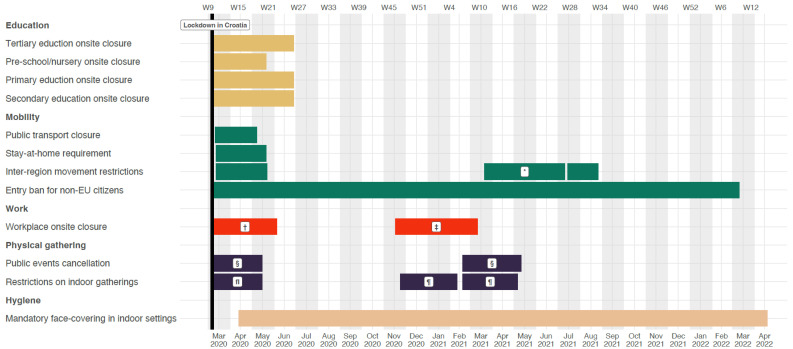
Timeline of implementation and relaxation of non-pharmaceutical interventions in response to the COVID-19 pandemic in Croatia. Five categories of non-pharmaceutical interventions were summarized: education, mobility, work, physical gathering, and hygiene from 16 Mar 2020 to 4 April 2022. Black vertical line indicates when lockdown was in Croatia on 16 March 2020. * Total border closure. † Closure of public spaces. ‡ Restaurants and cultural institutions closure. § Ban on holding all public events. fI Ban on gathering >5 persons in one place. ¶ Ban on gathering >10 persons in one place.

**Figure 2 viruses-14-02746-f002:**
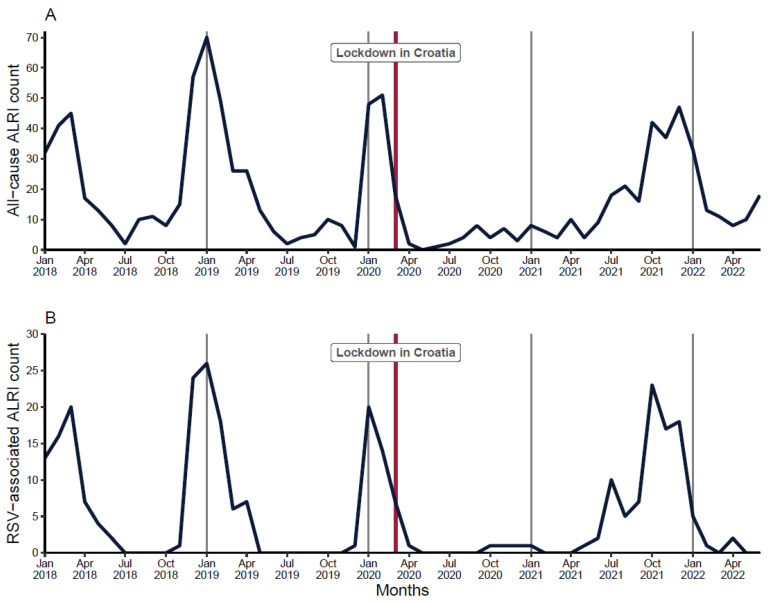
Month-by-month distribution of hospitalizations due to all-cause (panel **A**) and RSV-associated (panel **B**) ALRI. Black line: Number of hospitalizations. Red vertical line: Lockdown in Croatia on 16 Mar 2020.

**Figure 3 viruses-14-02746-f003:**
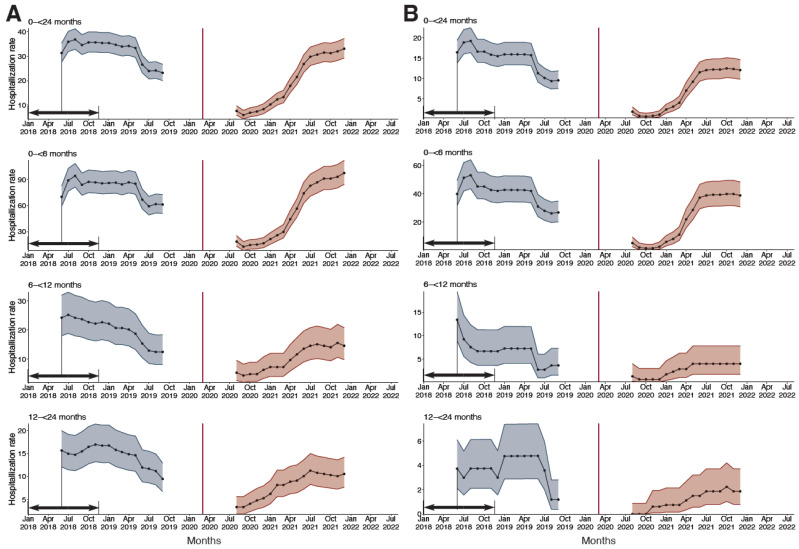
Change in 12-month hospitalization rates (per 1000) of all-cause and RSV-associated ALRI by age group, before and after the onset of the COVID-19 pandemic. (**A**) 12-month all-cause ALRI moving average hospitalization rate. (**B**) 12-month RSV-associated ALRI moving average hospitalization rate. Black arrows denote the time interval used for calculating the 12-month hospitalization rates. Red vertical line: lockdown in Croatia on 16 Mar 2020.

**Table 1 viruses-14-02746-t001:** Basic characteristics of study subjects.

	All-Cause ALRI (N = 942)			RSV-Associated ALRI (N = 282)		
	1 January 2018 to 16 March 2020 (N = 592)	17 March 2020 to 30 June 2022 (N = 350)	*p* Value	1 January 2018 to 16 March 2020 (N = 183)	17 March 2020 to 30 June 2022 (N = 99)	*p* Value
Sex, N (%)			0.539 ^a^			0.436 ^a^
Male	347 (58.6)	198 (56.6)		104 (56.8)	61 (61.6)	
Female	245 (41.4)	152 (43.4)		79 (43.1)	38 (38.4)	
Age group, N (%)			<0.001 ^a^			0.154 ^a^
0–<28 d	30 (5.7)	52 (14.8)		14 (7.7)	17 (17.2)	
28 d–<3 m	212 (35.8)	121 (34.6)		94 (51.4)	45 (45.5)	
3–<6 m	126 (21.3)	67 (19.1)		45 (24.6)	23 (23.2)	
6–<12 m	93 (15.7)	44 (12.6)		21 (11.5)	8 (8.1)	
12–<24 m	131 (22.1)	66 (18.9)		9 (4.9)	6 (6.1)	
Gestational age, N (%)			0.200 ^a^			0.723 ^a^
<28 w	10 (1.7)	5 (1.4)		2 (1.1)	0 (0.0)	
28–<32 w	10 (1.7)	7 (2.0)		3 (1.6)	1 (1.0)	
32–<37 w	48 (8.1)	43 (12.3)		22 (12.0)	13 (13.1)	
≥37 w	524 (88.5)	295 (84.3)		156 (85.2)	85 (85.9)	
Birth weight, median (IQR)	3470 (3060, 3820)	3420 (3000, 3737.5)	0.341 ^a^	3470 (2980, 3820)	3480 (3010, 3810)	0.931 ^b^
Birth height, median (IQR)	50.5 (49, 52)	50 (49, 51.8)	0.023 ^b^	50 (49, 52)	50 (48, 51)	0.261 ^b^
Comorbidity, N (%)						
Any	43 (7.3)	47 (13.4)	0.002 ^a^	10 (5.5)	6 (6.1)	0.836 ^c^
Congenital heart disease	16 (2.7)	25 (7.1)	0.001 ^c^	5 (2.7)	5 (5.1)	0.315 ^c^
Chronic pulmonary disease	15 (2.5)	15 (4.3)	0.139 ^c^	3 (1.6)	1 (1.0)	-
Neuromuscular disease	2 (0.3)	1 (0.3)	-	0 (0.0)	0 (0.0)	-
Immunodeficiency	1 (0.2)	0 (0.0)	-	1 (0.5)	0 (0.0)	-
Down syndrome	9 (1.5)	6 (1.7)	0.818 ^c^	1 (0.5)	0 (0.0)	-
Severity of ALRI, N (%)						
Severe	17 (2.9)	29 (8.3)	<0.001 ^a^	3 (1.6)	10 (10.1)	<0.001 ^c^
Very severe	14 (2.4)	14 (4.0)	0.153 ^a^	3 (1.6)	6 (6.1)	0.044 ^c^
Length of hospital stay in days, median (IQR)	5 (4, 7)	6 (4, 8)	<0.001 ^b^	6 (4, 8)	7 (5, 8)	0.039 ^b^

Note: ALRI, acute lower respiratory infection; RSV, respiratory syncytial virus. ^a^
*p*-value calculated by Chi-square test, ^b^
*p*-value calculated by Mann-Whitney U test, ^c^
*p*-value calculated by Fisher’s exact test.

**Table 2 viruses-14-02746-t002:** Multivariate logistic regression results of the risk factors of severe or very severe illness of all-cause and RSV-associated ALRI hospitalizations.

	Odds Ratio for All-Cause ALRI		Odds Ratio for RSV-Associated ALRI	
Variables	Estimate	95% CI	Estimate	95% CI
Male	0.94	0.57–1.55	0.71	0.28–1.81
Age				
0–<6 m	1	Reference	1	Reference
6–<12 m	1.40	0.70–2.67	2.38	0.61–7.75
12–<24 m	1.70	0.92–3.04	0.90	0.05–5.32
Preterm (<37 wGA)	0.46	0.21–1.07	1.08	0.25–5.95
Low birth weight (<2500 g)	0.69	0.26–1.73	2.04	0.38–9.52
Any comorbidities	5.02	2.56–9.60	0.58	0.03–3.58
Admitted during the COVID-19 pandemic	1.92	1.17–3.15	5.03	2.01–13.89

Note: ALRI, acute lower respiratory infection; RSV, respiratory syncytial virus; CI, confidence interval; m, months; w, week; GA, gestational age. Increased severity is defined as any severe or very severe ALRI.

## Data Availability

The data presented in this study are available on request from the corresponding author.

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
