# Peer review of "Changes following the Onset of the COVID-19 Pandemic in the Burden of Hospitalization for Respiratory Syncytial Virus Acute Lower Respiratory Infection in Children under Two Years: A Retrospective Study from Croatia"

_viruses, 2022, doi:10.3390/v14122746_

Round 1

Reviewer 1 Report

Dear editor,

Thank you for the opportunity to review this interesting manuscript describing the changes in RSV epidemiology in Croatia due to the COVID-19 pandemic. The manuscript clearly describes the number of hospitalizations for all cause ALRI and RSV associated ALRI in children from Jan 2018 untill June 2022 in one hospital in Croatia, and relate this to the COVID-19 non-pharmaceutical measures in place in Croatia. This study confirms the absence of RSV in the 2020/21 winter  in Croatia and the resurgence from April 2021 onwards. In addition, the authors have done several regression analyses to investigate differences in patient characteristics of children hospitalized before and after the pandemic.

More knowledge regarding the epidemiology and disease burden of RSV is important to support decisions regarding future immunization therapies that are currently in late stage clinical development. This study makes an important contribution to evaluate the burden of disease and epidemiology in Croatia.  

Comments/suggestions

I have one important comment regarding the conclusions the authors have drawn. They report in the discussion “the study demonstrated higher proportion of severe and very severe all-cause and RSV-associated ALRI admissions during the COVID-19 period, after adjusting for potential confounders such as age and comorbidities; it is unknown whether the observed increased severity will persist in the following RSV seasons”.

Although this text is technically correct, I think it is important to highlight that this increased severity can be caused because children are more severe ill (as suggested by the authors), but another possibility is that the health care seeking behavior and health care accessibility is changed due to the COVID-19 pandemic. I can imagine that parents are waiting longer to go to the hospital with their ill child because they are afraid a family member get infected with SARS-CoV-2, or they have had a negative SARS-CoV-2 test result and were reassured for a couple of days, and/or that the workload in hospitals has been increased due to COVID-19 and physicians need to be more critical in their decision whether or not to hospitalize a child. In these three situations the proportion of hospitalized children with severe illness is also increasing, but this does not automatically reflect an increased severity of children with an RSV infection in the population. I suggest the authors reflect on these three other possible explanations too. In addition, I suggest that authors are more carefull in reporting this increased severity statement in the conclusion and abstract.

The same goes for the conclusions regarding the age distribution of hospitalized children with all cause ALRI and RSV associated ALRI – the authors found no significant differences between the pre and post pandemic periods. Other studies found that more older children are infected with RSV in the post pandemic period. I would like to ask the authors to reflect if changes in health care seeking behavior/health care accessibility and RSV testing practices might have had an impact on the age distribution of hospitalized children in Croatia? Do the authors think there might be relation with the setting e.g. community/ primary care/hospital the study was conducted?

Minor comments:

Line 96: Heading “Testing for RSV”. Can authors add more information whether the included children were previously tested for SARS-CoV-2? And are testing policies changed in Croatia due to the COVID-19 pandemic?

Results: Can the authors indicate the percentage of children with All-cause ALRI and RSV-associated ALRI are tested for RSV? Are there differences between the pre and postpandemic period in testing practices? I would suggest to explain this shortly in the main text and not only in the supplementary table.

Table 2: Title: “Multivariate logistic regression results of the risk factors for increased severity of all-cause and RSV-associated ALRI hospitalizations”.

I suggest to replace ‘increased severity’ into ‘severe or very severe illness’. This is in line with the analyses performed namely logistic regression analysis.

Table 2: I expect the length of hospital stay is in days, I suggest to add this to the table.

Table S1: Number of ALRI patients tested for RSV is reported as 695 cases (left part) and 708 (right part). Can you explain this difference?

Supplemental files S2, S3, S4: It is not clear that the estimates are reflecting odds ratios. I suggest to use the same reporting as in table 2.

Author Response

Thank you for taking the time to review this manuscript.

Please see the attachment for the point-by-point response.

Reviewer 2 Report

Overall well written report. Methodologically there was nothing to complain. There are many similar papers and thus the editor should decide about the publication priority, but overall there were no issues in the reporting. 

Author Response

Thank you for your comment.